# Self-Healing Potential and Phase Evolution Characterization of Ternary Cement Blends

**DOI:** 10.3390/ma13112543

**Published:** 2020-06-03

**Authors:** Mojtaba Mohammadi, Carol Youssef-Namnoum, Maxime Robira, Benoit Hilloulin

**Affiliations:** 1Institut de Recherche en Génie Civil et Mécanique (GeM), UMR 6183, Ecole Centrale de Nantes—Université de Nantes—CNRS, 1 rue de la Noë, 44321 Nantes, France; mohammadi.mojtaba1991@gmail.com (M.M.); carol.youssef-namnoum@ec-nantes.fr (C.Y.-N.); robira@arronax-nantes.fr (M.R.); 2Groupe d’Intérêt Public ARRONAX, CEDEX, 44817 Saint-Herblain, France

**Keywords:** autogenous self-healing, ternary cement blends, supplementary cementitious materials (SCMs), phase evolution, healing potential, X-ray diffraction, thermogravimetric analysis

## Abstract

The autogenous self-healing of cementitious material micro-cracks might lead to the service-life extension of structures. However, most of its aspects are still unknown. This paper investigates the self-healing capacity of ternary cement blends including metakaolin (MK), ground granulated blast-furnace slag (BFS), limestone (LS), and siliceous filler (F). Morphology and healing precipitation patterns were studied through the optical microscopy of artificial micro-cracks, global healing product mass monitoring, and XRD and TGA used to identify and quantify mineral formation. The self-healing potential index is introduced based on the mass measurements. It was found that the formulation containing 10% MK presented the highest healing potential at an early age (<28 days), while the formulations containing 20% BFS with 10% LS/F showed a higher healing potential at an older age (cracked after 28 days of curing). Calcite, C-S-H, and portlandite were found to be the main healing products alongside specific formulation-dependent compounds, and it was observed that the calcite’s relative quantity generally increased with time. Finally, the evolution of the self-healing product phases was accurately monitored through XRD and TGA measurements.

## 1. Introduction

Global warming, due to the increase of greenhouse gas emissions, is currently an important issue, and the cement industry produces a huge amount of carbon dioxide in the cement production process. However, concrete—due to its affordability, its good compressive strength, and the local availability of raw materials—is used on a large scale [1]. The ongoing increase of construction, especially in East Asia and Middle East, is increasing the demand of cement. Therefore, considering its mass, cement is the largest man-made product on the earth [2], and important amounts of natural resources are consumed in its use [3]. Hence, replacing a part of cement with supplementary cementitious materials can decrease its carbon dioxide footprint and could have a positive effect on the natural ability of concrete to heal after cracking due to direct or indirect loading [4]. Moreover, using supplementary cementitious materials could decrease the consumption of natural resources and expensive compounds like cement, thus reducing the cost of concrete [5].

Cement-based materials like concrete are chemical-active media; hence, they are vulnerable to be attacked by harmful chemical materials. Self-healing is a short-term solution to decrease the permeability and to increase the durability of cement composite structures [6,7]. Self-healing could result in mechanical regains at the same time [8,9]. According to the healing mechanism, there are two main strategies for self-healing: autogenous self-healing and autonomous self-healing. Autonomous self-healing is caused by manually adding a specific agent to a cementitious matrix before or after casting [10]. The specific agent can be added by embedding capsules charged with corresponding healing agents [11], hollow fibers, shape-memory materials, and superabsorbent polymers (SAPs) [2], or by applying special bacteria on the crack surface or inside the matrix. Autogenous self-healing refers to the natural capacity of cementitious materials to heal by themselves without human intervention. Autogenous self-healing is mainly based on two chemical processes: carbonation and ongoing hydration. Carbonation is a precipitation process of calcium carbonate in a crack due to the presence of dissolved carbon dioxide oriented from the atmosphere, water, and calcium from the matrix (usually from the dissolution of portlandite) [3]. The crystallization of calcite is the major self-healing mechanism of mature concrete [12]. Ongoing hydration could be a result of the hydration of unreacted particles, latent hydraulics, or the leaching of the ions from the depth of a matrix to the surface of cracks. Ongoing hydration in cracks is different from ongoing hydration in a bulk. For example, in cracks, there is more water available and more space available [13]. Therefore, the formation of crystalline and hydration products could be more expected in cracks than in a matrix, and this may also influence the nucleation and growth processes of healing products [8].

Because of the activity of Portland clinker particles, there are less unreacted particles available for a second hydration process. Hence, applying latent hydraulic materials as supplementary cementitious materials (SCMs) that have lower reaction rates than clinker particles can increase the time of the availability of unreacted particles. It could therefore increase the potential self-healing capacity of cementitious materials. It has been shown that replacing cement by blast furnace slag or fly ash in a binary binder can enhance further hydration [5]. During the first weeks of water curing, binder-containing SCMs show a rapid self-healing evolution [14]. Cracks with widths in the range of 100–200 µm are closed, and mechanical recovery is obtained [5,14]. The study of binary and ternary binders appears to be a promising research path [15] because of the environmental benefits and potential positive influence on the self-healing process.

Cementitious materials, including SCMs, have also been found to be very effective from an environmental point of view with mechanical properties and durability comparable to (sometimes higher than) compositions with pure cement when carefully selected. Among SCMs, limestone, blast furnace slag, and metakaolin have been proved to be particularly effective. Their main characteristics can be described as follows:Limestone is one of the most available materials in the Earth’s crust (10% of sedimentary rocks). Limestone has recently become widely used as part of Portland cement and standardized in Canada (CSA A3000-08), Europe (CEM II), and the USA (ASTM C595-12), with substitution rates around 15% [16]. It affects the hydration of cement by reacting with tricalcium aluminate (C_3_A). This reaction leads to the formation of the carboaluminate. Hence, the addition of the supplementary materials that contain alumina allows for more limestone to be added [17].Blast furnace slag is a by-product material that comes from the iron and steel industry. It consists mostly of silica and burnt lime (CaO). Its hydration is similar to that of clinker, but it produces lower amounts of CH and C-S-H [18]. The main hydration products of blast furnace slag are C-S-H gel; CH; sulfur-aluminate hydrate phases like ettringite (AFt) and monosulfoaluminate (AFm); and Mg and Al-rich hydroxide phases [4].Metakaolin is a manufactured material from clay with excellent pozzolanic properties. It mainly consists of silica, alumina, and (eventually) quartz, and it has a higher specific surface area than cement [17,19]. It can increase the hydration rate and the kinetics of C-S-H formation. In addition, after dissolution, it provides additional Si, or it can act as a nucleation site for C-S-H formation. This can result in the production of monocarboaluminate and hemicarboaluminate.

In this research, the influence of ternary binders on the self-healing potential of cementitious materials was studied. For the first time, the kinetics of healing were monitored using mass measurements to find out the possible relations between hydration and self-healing. A new parameter that can classify different compositions based on their ability to heal and that is useful to measure the kinetics of hydration during different periods of healing is defined. Finally, a novel assessment of the phase evolution of self-healing products over time is presented through the comparison of XRD and TGA tests.

## 2. Materials and Methods

### 2.1. Materials

In this study, the materials used were ordinary Portland cement (PC) (CEM I 52.5 N), silica sand (0/2), siliceous filler (F) as a neutral material with a d50 equal to 34 (μm), and supplementary cementitious materials (SCMs) such as blast furnace slag (BFS), metakaolin (MK) with a d50 equal to 15 (μm), and limestone (LS) with a d50 equal to 8.5 (μm). The chemical and physical properties of the binder materials are shown in Table 1.

Eight different mortar mixtures with a water to cement ratio (W/C) of 0.5 were prepared according to the proportions given in Table 2. All mixtures were designed with the same paste volume (5 L), where half of the mortar volume consisted of sand and the other half consisted of binder. The binder phase included 70% of Portland cement and 30% of the cement that was replaced by SCMs or siliceous filler.

The mixture denoted by F30 represents the reference mixture in which SCMs were not added. F30 was made with 70% of PC and 30% of neutral material F to represent the same content of cement for all mixtures. Then, a combined mixture of SCMs partially replaced the filler by up to 30% vol. In order to study the combined effect of two SCMs, the mortar mixtures were prepared in the following ways. First, MK20–X10 specimens were made with 20% of metakaolin and 10% of X, where X = {F, BFS and LS}. Second, BFS20–Y10 specimens were made with 20% of BFS and 10% of Y, where Y = {F, MK and LS}. 

### 2.2. Sample Preparations

All the mixes were cast in steel greased molds sized 40 × 40 × 160 mm^3^. The mixture was vibrated into two layers on a horizontal vibrating table for 20 s every time to minimize the amount of occluded air. The specimens were demolded after 24 h and cured in tap water in a room with a controlled temperature of 20 °C and a relative humidity larger than 90%.

Compressive tests were performed on three 40 × 40 × 40 mm^3^ cubic specimens at 2, 7, and 28 days. These specimens were cut using a circular saw. The average values of the mechanical properties were calculated.

### 2.3. Artificial Cracks Preparation and Healing Conditions

In order to investigate the self-healing process of materials containing SCMs, the reaction products formed in artificial cracks were characterized and the amount of healing products was quantified. A special procedure was followed [7]: an artificial planar gap was made by cutting the 40 × 40 × 160 mm^3^ sample into 16 slices, each slice having a thickness of around 10 mm. After that, the slices were pressed together by using a plastic clamp to obtain a crack width of less than 150 µm. The specimens were then cured in tap water for various healing times (Figure 1). By using tap water, the simulation was closer to reality and avoided the leaching of healing products. Similarly to the healing process occurring in real cracks, new reaction products were precipitated inside the artificial planar gaps between two slices.

For each mortar mixture mentioned before, two series of specimen were prepared. As shown in Figure 2, series 1 specimens were artificially cracked after two days of curing, while series 2 specimens were cracked after 28 days of curing. To evaluate the influence of the healing period, the specimens prepared in every series were subjected to 7, 14, and 28 days of healing. For instance, mixture MK20–BFS10 2C8-28H was cracked at 28 days and then subjected to 28 days of healing.

### 2.4. Healing Product Conditioning and Analysis

After the healing period, a micrograph of the surface of the slices was directly taken in order to obtain volume information on the development of healing products between slices. Then, the newly formed products were scratched off from 64 surfaces by means of medium P80 sand paper for each healing duration. The efficiency of scratching was controlled by microscopic observation to avoid scratching the cement paste powder. The healing product mass obtained from each mixture was subsequently measured by an analytical balance.

All of the formulations based on their specific compositions showed a certain level of autogenous self-healing. A ‘healing potential’ is thus defined as the ratio, intrinsic to each formulation, between the mass of the collected powder divided by the total crack surface:(1)$$\mathrm{Healing}\;\mathrm{potential}=\frac{m}{s\;\times 2n\;\times \;b}$$
where m is the quantity of measured powders (mg), s is the surface of slices (cm^2^), n is the number of slices obtained by cutting one 40 × 40 × 160 mm^3^ specimen, and b is the number of specimens prepared by the mixture. The healing potential unit is mg/cm^2^. Between 5% and 10% of handling errors were considered when calculating total surface and when comparing amount of healing products between different formulations.

In order to get more information on the chemical and mineral structure of healing products, the following techniques were used:XRD was carried out with a Bruker D8 phaser instrument (Karlsruhe, Germany). The tests were conducted with the continuous scanning of a detector covering an angular range between 10° and 80° with a step rate of 0.0195 for 2θ. The collected data were analyzed using the Match! software (version 3, Crystal Impact, Bonn, Germany).TGA was performed with a Netzsch STA 449 F3 Perseus device (NETZSCH-Gerätebau GmbH, Selb, Germany) in a nitrogen protection of 0.6 bar pressure. The powders were heated from 27 to 1100 °C at a heating rate of 20 °C/min in a ceramic crucible and with a single ramp.

Crack healing is not constant through time. Therefore, the following was introduced to quantify the healing kinetics at different healing time steps:(2)$$\mathrm{Kinetics}\;\mathrm{of}\;\mathrm{healing}=\frac{m_{\mathrm{xC}-\mathrm{yH}}}{t}$$
where $m_{\mathrm{xC}-\mathrm{yH}}$ is the collected healing mass (mg) from the crack surface after x days of curing (C) and y days of the healing (H), and t is the healing period duration (day).

## 3. Results and Discussion

### 3.1. Morphology of Healing Products on the Crack Surfaces

During the self-healing period, the former planar crack surfaces were covered by a layer of precipitates. Different types of morphologies that can be classified into two major groups were detected: the crystal-like precipitates and the gel-like precipitates, both of them corresponding well with previous results [13]. The crystals had similar forms for all formulations. As illustrated in Figure 3, crystal-like precipitates could be classified in cabbage-like (Figure 3d), needle-like (Figure 3e), and irregular-shape (Figure 3f) types. The cabbage-like crystals were found to be more condensed, thus leading to a greater filling percentage of the cracks. Conversely, the crack filling percentage due to needle-like products was smaller. 

Observing internal crack surfaces with an optical microscope and focusing on different morphologies and their position revealed some interesting information. Figure 3a shows pictures of the internal crack surfaces from crack edges to center. The crystals that were formed near the crack mouth (a distance of around 1200–1400 μm to the crack lips) were mostly composed of non-dense crystals that were calcite precipitates [8]. Then, further inside the crack, these crystals became denser (as is shown by a yellow arrow in Figure 3a) and cabbage-like crystals that could be C-S-H compounds were observed [13]. Moving toward the center, the healing products were more gel-like. Finally at the center, the healing products were crystal-like and condensed on most of the cases.

It should be noted that besides the density of precipitates, the ability to make bridges is another important factor that helps mechanical regain. Figure 4 shows the development of crystals at the mouth of cracks in relation to their type. The orange-colored arrow in Figure 4b depicts a crystal bridge, where the green-colored arrow depicts a growing crystal connecting to the crystals that were growing from the other side of the crack. However, for the cracks with the crystals that were not enough grown to form a bridge (like what is shown by the orange arrow in Figure 4a), the crack could not be filled. In this case, the cracks were not closed, but they were narrowed, thus leading to a permeability decrease.

### 3.2. Self-Healing Mass Production

#### 3.2.1. Measurement of Matrix Healing Potential

After collecting healing products from crack surfaces and their quantification, it was found that the healing quantity varied between different formulations and even in between different healing durations. Using Equation (2), the healing potential for different formulations and different healing duration was obtained. Table 3 represents the healing potential of the mortar formulations when considering the healing period and curing time. Amongst the values from series 1 (specimens cracked at two days), the samples with a longer self-healing duration (28 days) generally showed a higher healing potential compared the same formulations with a shorter healing duration. Indeed, the MK20-F10 formulation exhibited the highest healing potential after 28 days of healing. This highlights the beneficial influence of the moderate metakaolin contents on the self-healing potential after an early age cracking. Within the series 2 specimens cracked at 28 days, the formulations with 20% of BFS20 and 10% of LS/F showed the better healing potentials. Moreover, Table 3 demonstrates the higher healing potential of specimens from series 1 as compared to specimens from series 2. However, even after 28 days of curing, the specimens showed some noticeable healing potential, which demonstrated the self-healing potential of mature mortars including SCMs.

#### 3.2.2. Kinetics of Healing Formation

Self-healing kinetics was not constant through time, as illustrated Figure 5, which shows the kinetics of self-healing for specimens sliced after two days. Accordingly, a general decrease in healing kinetics was observed during the 28 days of healing, as observed in previous studies [14]. The results confirmed a higher capacity to form healing products during the first seven days of healing. This could have been a result of the availability of unreacted particles and an abundance of ions due to leaching from the depth of the matrix to the cracks [13,20,21], while, at later ages, the access to the deeper unreacted particles and ions could be denied due to the presence of precipitates and the porosity decrease. The formula with MK20–LS10 showed fast kinetics even after 14 days of healing. Results from the specimens sliced after 28 days of curing showed very low speeds.

### 3.3. Characterization of Self-Healing Products

#### 3.3.1. X-ray Diffraction

##### Second Hydration, Self-Healing Products and XRD Patterns

The phases and minerals of the self-healing products and the corresponding reference specimens from the mortar matrix were identified for all the formulations. Figure 6 presents a typical XRD pattern wherein every specific set of peaks represents a particular phase; Table 4 summarizes the results. Table 4 indicates that healing product patterns included a wide range of minerals and phases (calcite, portlandite, ettringite, gibbsite, C-(A)-S-H, hemicarboaluminates (Hc), monocarbonate (Mc), brucite, kuzelite, siliceous hydrogaret, C_4_AH_13_, aragonite, vaterite, Si_3_O_8_(OH)_2_, and belite). One piece of interesting information is the presence of belite (C_2_S) among all of the reference XRD patterns. Belite is not a healing product, but its detection after 28 days when the mortar reached its maximum compressive strength due to the maximum hydration of clinker particles indicated a preserved healing potential in the lifetime of the mortar.

Self-healing products are different from the original hydration products precipitating in the mortar matrix. Using a reference pattern, the second hydration phases during self-healing could be compared with the main hydration phases. Pattern B on Figure 6 represents the XRD pattern of the specimen of series 1 that was healed (H) for 28 days (2C-28H) and patterns C and D represent the XRD patterns attributed to series 2 that, respectively, healed for 7 days (28C-7H), and 28 days (28C-28H). There were noticeable differences in the number and position of peaks among the patterns. It can be interpreted that every specific formulation and healing period had its particular set of minerals. Moreover, in most cases, more peaks were found in the reference than in the healing products, thus indicating that the phases in the matrix were more diverse than the healing products phases.

Self-healing products exhibited similarities over the various ternary blends, as can be observed in Table 4. Some phases were available in all of the healing products and could be considered the major healing products, like calcite [8], portlandite, C-(A)-S-H, and gibbsite Al(OH)_3_. On the other side, some peaks were more specific to some formulations such as aragonite (CaCO_3_ polymorph) that may have had a positive influence on healing due to their geometry, ettringite, Hc, and Mc that were interestingly found in almost all of the ternary formulations, kuzelite found in the ternary mixes containing metakaolin and blast furnace slag, and brucite found in formulations containing blast furnace slag.

The origin of the healing products could be assessed based on the various chemical reactions reported in the literature related to ternary blends. The presence of calcite in the matrix could have been due to the presence of unreacted calcite particles from limestone, and they could have been the result of the reaction between Ca^2+^ ions (one of the needed ions for calcite formation) that were largely available due to leakage through the matrix or the dissolution of calcium hydroxide and/or lime under highly alkali crack solutions and the presence of carbon dioxide (dry or soluble in water [22,23]). Additionally, based on the Glukhovsky model describing destruction–coagulation–condensation–crystallization [24], the unreacted particles of the supplementary materials containing silica and alumina constituted a silica chain of Si-O-Si and an alumina chain of Al-O-Al, thus leading to the formation of nucleation sites that promoted the development of C-A-S-H [10]. In addition, the available nucleation sites due to the presence of metakaolin resulted in the formation of C-S-H with a denser structure that made the media more resistant to a chemical attack by chlorides [25]. The kuzelite phase (Ca_4_Al_2_(OH)_12_(SO_4_)·6(H_2_O)) was only found in healing powders of the formulations containing both MK and BFS. Kuzelite was the result of the reaction between amorphous, oxide-free silicone in metakaolin and sulfate SO_4_^2−^ ions in cement components [10,25]. The healing powder patterns corresponding to the formulations MK20–BFS10, BFS20–MK10, and MK20–F10 contained a set of peaks related to brucite (MgO), a hydrate that precipitates slowly as a result of the slow hydration of MgO particles in clinkers or BFS. In addition, these formulas only showed some contents of hemicarboaluminate (Hc) in their healing products. Hc formation was due to the addition of MK and BFS, which significantly increased the amount of alumina in the mix; therefore, the reaction of alumina and produced calcite, which resulted in the formation of hemicarboaluminate [17]. Then, due to the presence of hemicarboaluminate, the reaction of alumina with calcium carbonate and portlandite resulted in the formation monocarboaluminate. The C_4_AH_13_ healing phase was found in formula with MK/BFS and MK20–FS10, while their reference pattern showed no corresponding peak. This could have been the result of the presence of gypsum particles in the matrix paste, while the presence of lime CaO and absence of gypsum resulted in the formation of C_4_AH_13_ phases [26]. Furthermore, a reaction between portlandite and C_3_A particles [10] could have resulted in the formation of C_4_AH_13_ phases. Finally, siliceous hydrogaret (Ca_3_(Al_x_Fe_1−x_)_2_(SiO_4_)_Y_(OH)_12_) [25] with an extremely slow formation reaction at room temperature in an aluminum–iron intermix was specifically found in healing products of the formulation with 30% of siliceous filler.

##### Comparison of XRD Patterns of Different Formulations

After the identification of the mineral phases in the self-healing products of the different mixtures, it was possible to compare the formulations to eventually select among the SCMs. According to Table 4, in the formulation containing only siliceous filler (F), which was expected to be a chemically neutral material (after 28 days of healing), the healing powder contained the common products plus C-A-S-H while the removal of F and the addition of 20% of MK and 10% of BFS resulted in the disappearance of C-A-S-H and the production of kuzelite (Ca_4_Al_2_(OH)_12_(So_4_)·6H_2_O) and aragonite. It can be interpreted that C-A-S-H was not produced or was transformed into new phases. After adding MK and BFS to the binder, the amount of alumina and silica noticeably increased, and the amount of aluminum and silicon ions increased. This increase led to the formation of the phases like hemicarboaluminate and monocarboaluminate or changes in the quantity of the phases, such as increases in C-(A)-S-H or decreases of calcite during the formation of monocarboaluminate [17].

In addition, regarding ettringite as a healing product, it was found that MK had a good effect on its formation. Indeed, the formulation incorporating 20% of metakaolin and 10% of BFS/F incorporated ettringite among its healing phases for the whole period of healing. Moreover, formulations with 10% of MK showed ettringite for 80% of the healing periods. In another view, monocarboaluminate and hemicarboaluminate were formed instead of monosulfoaluminate, thus leaving a greater portion of sulfate [17]. Thereforemore ettringite was formed in the presence of Al^3+^ and Ca^2+^ as more sulfate became available.

Finally, according to Table 4, kuzelite was formed in mixtures including MK/BFS and silica- and calcite-based supplementary materials like BFS, and LS was needed to have Hc and Mc within the healing phases.

##### Phase Evolution of the Healing Products with Time

A detailed analysis of the XRD patterns suggested the evolution of the healing products with time. Some peaks appeared in a period of self-healing and disappeared in another period. Considering Table 4, brucite was a phase formed after 28 days of healing, while no peaks were found at the beginning. In fact, at the beginning of healing, magnesium ions were likely to participate in the formation of Mg-Al in hydrotalcite (Ht); then, after time and the hydration of MgO [27], new C-S-H were formed and freed magnesium ions that could be precipitated in a hydroxide form or brucite.

Ettringite is one of the phases that reflected changes during time. The formation of ettringite was due to the rapid reaction of calcium ions with sulfate ions, and studies have indicated that ettringite can be converted to monosulfoaluminate or, when converted with calcite, turned into hemicarboaluminate [10]. These formation conversion cycles could corroborate the appearance and disappearance of the ettringite phase during the healing periods. Additionally, for the formula with 20% of BFS and 10% of MK that cracked at 28 days and healed for 28 days, Table 4 shows the disappearance of both ettringite and hemicarboaluminate. However, the phase corresponding to monocarboaluminate existed, indicating the late formation of monocarboaluminate by hemicarboaluminate and the use of ettringite to produce more hemicarboaluminate.

Furthermore, C-S-H peaks experienced some appearances and disappearances. The absence of C-S-H after the shortest healing periods in some formulations could have been due to its slow development in the crack [8], whilst the participation of C-S-H in forming new products like its reaction with hemicarboaluminate to form stratlingite and calcium carbonate phases could be considered of the mechanisms that affects the quantities of C-S-H and hemicarboaluminate at the same time.

Finally it was found that C_4_AH_13_, which has a structure similar to monosulfate, was formed at ambient temperature at an early age—specifically in formulations containing lime or BFS [26]. OH-AFm was actually precipitated at an early age and then transformed into C_3_AH_6_, which is a more stable phase of the lime–silica–hydrate system [25].

#### 3.3.2. Thermogravimetric Analysis

##### Phase Identification through TGA Results

The phases and minerals of all of the formulations were identified and quantified using TGA. Figure 7 depicts a typical TGA result obtained on the sample incorporating 20% of MK and 10% of BFS sliced after two days and healed for 28 days. The weight loss of the homogeneous healing powder at a temperature of around 110 °C can be attributed to the water loss of C-S-H and ettringite, and the one between 140 and 180 °C can be attributed to carboaluminate hydrates [28]. The third drop on the derivative thermogravimetry (DTG) profile at a temperature of about 300 °C can be attributed to the decomposition of C_3_AH_6_ [25]. At a temperature of around 450 °C, there was a weight loss assigned to calcium hydroxide; finally, the last peak with a remarkable drop corresponded to the decomposition of calcite at a temperature range from 600 to 880 °C [14]. For all the formulations, a large drop due to calcite decomposition was observable, while the amplitude of the drop due to portlandite evolved with the healing duration. Hence, the TGA results had a good correlation with XRD output and validated its results.

##### Quantification of the Healing Product Phases

Table 5 shows the quantity of phases measured in the formulations. First, the quantity of the phases over the total sample quantity was measured, and then the relative percentage of the phases over the total amount of detected phases was calculated. According to the table, the amount of the C-S-H gel in the matrix powder (reference specimen) was much higher than in the healing powder. Conversely, the calcite quantity was smaller in the matrix than in the healing products. 

An average of 2.3% of portlandite out of the total healing mass was measured, and this amount was relatively independent of the formulation. In addition, some healing powders exhibited a slightly higher portlandite content than the corresponding reference matrix, which agreed well with previous numerical calculations [8]. C-S-H gel in the healing powder amounted to around 1.5–3.5% of the total mass with an average of 2.3%. A relative quantity of 8–20% of C-S-H showed a decrease due to ongoing healing.

Calcite content varied from 11.5% to 22%, showing noticeable changes regarding the formulation. In total, more than 60% of the detected healing products were classified as calcite, and this relative proportion increased with the healing time. Therefore, even though the calcite quantity was mainly dependent on the formulation, it was found to be the major healing product in ternary formulations even after cracking at an early age (two days). Additionally, calcite evolution (like its reaction), with available alumina results in hemicarboaluminate production [17] because a new phase can be considered as one of the reasons of fluctuation in calcite quantity in different formulations.

Finally, the presence of C_3_AH_6_ was expected in the formulation containing C_4_AH_13_ among its hydrates. While XRD peaks corresponding to C_3_AH_6_ were not found, some TGA graphs highlighted its presence. The formulations containing BFS showed C_3_AH_6_ phases in their TGA patterns, and the same formula showed C_4_AH_13_ in their XRD patterns, which meant that both of the phases could be available at the same time in a media [25]. On the other hand, some samples contained only C_4_AH_13_, which was not transformed and produced no more C_3_AH_6_ that was more stable. However, this stable form of the lime–alumina–hydrate system could become unstable due to its strong carbonation sensitivity.

## 4. Conclusions and Perspectives

In this paper, the autogenous self-healing phenomenon of micro-cracks in ternary cementitious materials was qualified and quantified using mass monitoring, X-ray diffraction, thermogravimetry analysis, and optical microscopy. The following conclusions can be drawn:Two categories of the self-healing morphology—crystal-like and gel-like—were observed on artificially cracked ternary cement blends through an optical microscope in agreement with previous research on pure cement pastes. In addition, in most of the micro-cracks, the crack closure happened at a depth of 1200–1400 μm from the external edges.The healing potential ratio was introduced based on the mass monitoring of the healing products. It represents the self-healing production of a square centimeter of a crack surface.The formulations containing 10% of metakaolin presented the highest healing potential among all the formulation of the first series (cracked after two days of curing), therefore, indicating its beneficial influence on ternary binder healing potential. Besides, the presence of 20% blast furnace slag in the formulations showed a noticeable healing potential that highlighted the advantageous performance of BFS in a ternary binder. Eventually, the self-healing kinetics showed a decreasing behavior through time; after passing its early age with a high rate of productivity, it reached its lowest amount after 28 days.Based on XRD and TGA measurements, calcite was found to be the main healing product along with C-S-H and portlandite. Other healing products such as C-A-S-H, monocarboaluminate, hemicarboaluminate, C_4_AH_13_, C_3_AH_6_, belite, SI_3_O_8_(OH)_2_, and brucite were formed depending on the formulation.

## Figures and Tables

**Figure 1 materials-13-02543-f001:**
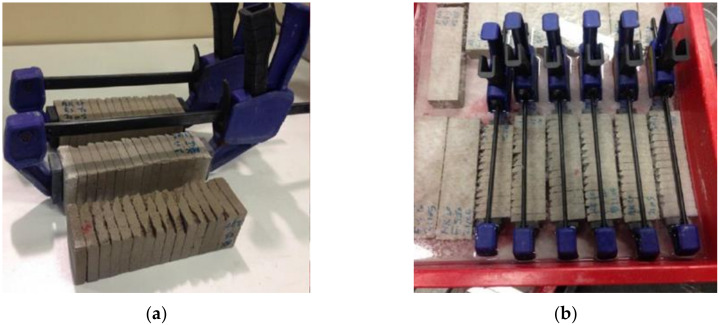
Experimental setup for artificial crack preparation: (**a**) Photos of slices cut and pressed together using a clump; (**b**) immersion of the pressed slices in tap water.

**Figure 2 materials-13-02543-f002:**
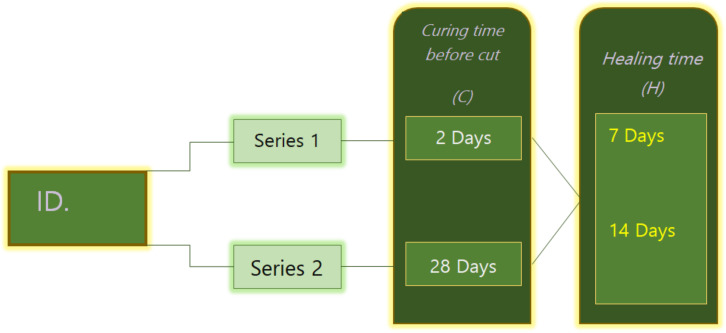
Program of cracking age and healing time of all mix design.

**Figure 3 materials-13-02543-f003:**
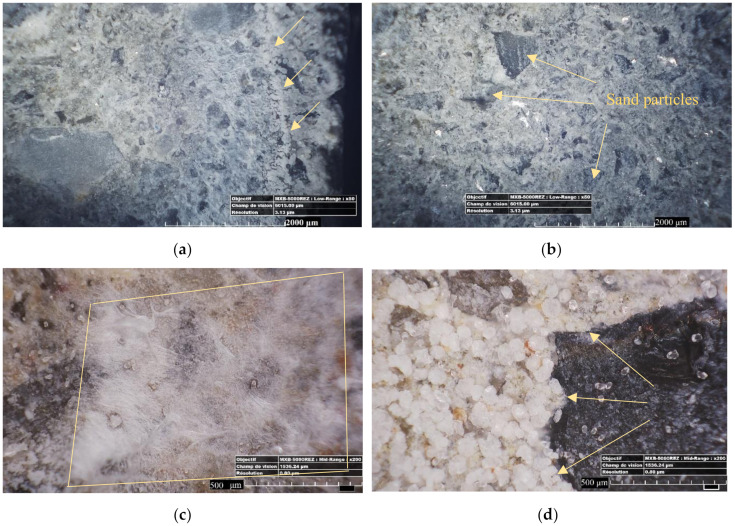
Various types of self-healing products deposited on the cracks faces: (**a**) a combination of the gel-like, crystal-like healing, and irregular shaped precipitates; (**b**) flake-like precipitates; (**c**) gel-like or fiber-like precipitates; (**d**) rounded cabbage-like crystals; (**e**) needle-like crystals; and (**f**) precipitates with irregular shape.

**Figure 4 materials-13-02543-f004:**
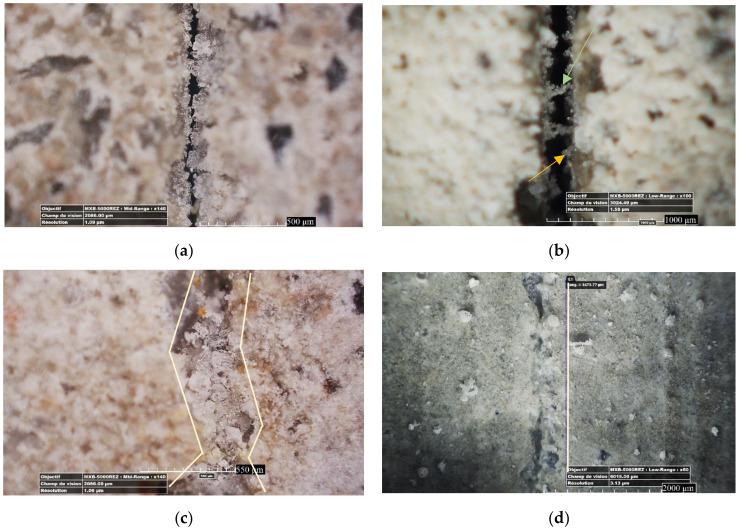
Optical observation of crack self-healing products from the crack surface: (**a**) MK20–LS10 at 36 days of age; (**b**) crack mouth is narrowed by crystal grows; (**c**) MK20–LS10 at 57 days of age; (**d**) BFS20–F10 at 7 days of age.

**Figure 5 materials-13-02543-f005:**
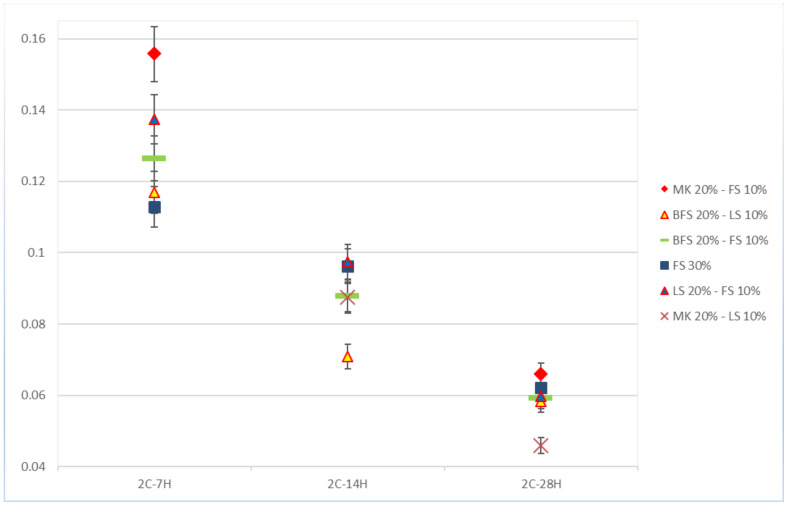
Self-healing kinetics of mortar samples from series 2. The kinetics attributed to MK20–LS10 after 7 days of healing is excluded from the graph because of the insufficient method of collecting healing products used for this specimen. The same applies to MK20–F10 after 14 days of healing.

**Figure 6 materials-13-02543-f006:**
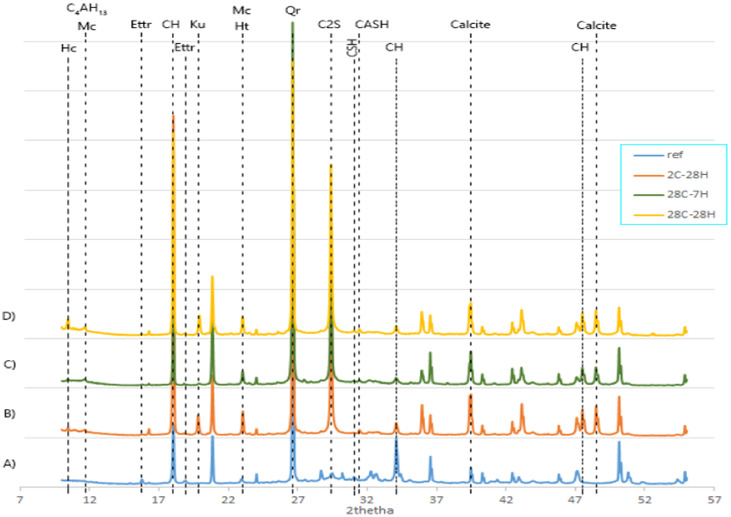
XRD patterns associated with the BFS20–MK10 formulation wherein pattern (**A**) corresponds to the matrix that was cured for 2 days, pattern (**B**) corresponds to the healing powder of the specimen sliced at 2 days and cured for 28 days, pattern (**C**) corresponds to the healing powder of the specimen sliced at 28 days and cured for 7 days, and pattern (**D**) corresponds to the healing powder of the specimen sliced at 28 days and cured for 28 days.

**Figure 7 materials-13-02543-f007:**
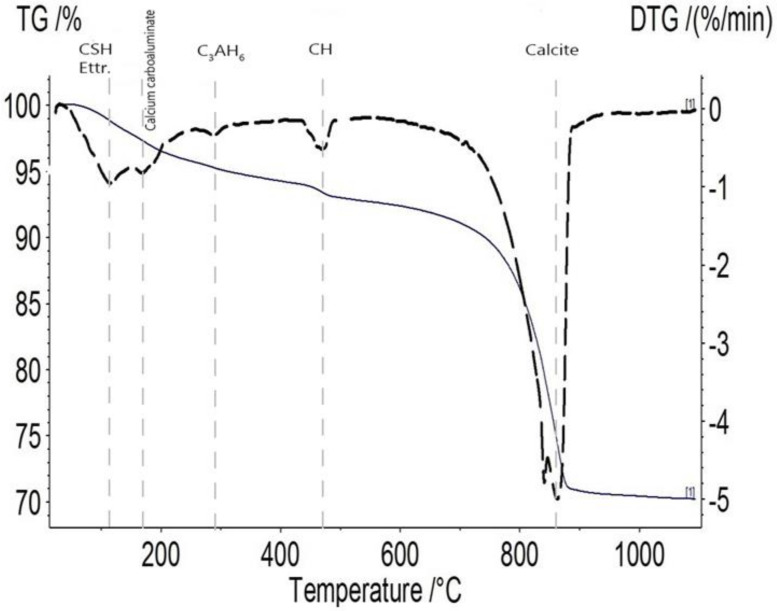
TG/DTG profile of the healing product belonged to 2C-28H healing products of MK20–BFS10 cracked at 2 days and healed during 28 days. Definitions: monosulfoaluminate (Ms), portlandite (CH), and ettringite (Ettr.).

**Table 1 materials-13-02543-t001:** Chemical compositions of Portland cement, supplementary cementitious materials (SCMs), and filler (mass %). MK: metakaolin; BFS: blast furnace slag; LS: limestone; and F: siliceous filler.

	PC	MK	BFS	LS	F
C_3_S	69	-	-	-	-
C_2_S	9	-	-	-	-
C_3_A	9	-	-	-	-
CaO	63	0.3	43.8	-	0.003
CaCO_3_	-	-	-	98.76	-
Al_2_O_3_	4.5	55	10.2	-	0.0048
SiO_2_	19.6	40	37.7	-	99.1
Fe_2_O_3_	2.3	1.4	0.6	-	0.0003

**Table 2 materials-13-02543-t002:** Mortar mix designs (kg/m^3^). MK20–X10 specimens were made with 20% of metakaolin and 10% of X, where X = {F, BFS and LS}. Second, BFS20–Y10 specimens were made with 20% of BFS and 10% of Y, where Y = {F, MK and LS}.

Formulation	W/C	PC	MK	BFS	LS	F	Sand
MK20–F10	0.5	828	180	-	-	100	1332
MK20–BFS10	0.5	828	180	30.5	-	-	1332
MK20–LS10	0.5	828	180	-	103	-	1332
BFS20–MK10	0.5	828	90	61	-	-	1332
BFS20–LS10	0.5	828	-	61	103	-	1332
BFS20–F10_3_	0.5	828	-	61	-	100	1332
LS20–F10	0.5	828	-	-	205	100	1332
F30 ^1^	0.5	828	-	-	-	300	1332

^1^ Reference formulation.

**Table 3 materials-13-02543-t003:** Healing potentials of mortar formulations for various curing (C) and healing (H) durations (mg/cm²). ’*’ indicate specimens scratched with an inefficient method.

Formulation	Series 1		Series 2	
	2C-7H	2C-28H	28C-7H	28C-28H
MK20–F10	1.174	1.985	0.634	1.083
MK20–BFS10	0.182 *	1.201	0.882	1.072
MK20–LS10	0.244 *	1.383	1.168	1.032
BFS20–MK10	0.273 *	2.044	0.956	0.914
BFS20–LS10	0.882	1.755	1.964	1.205
BFS20–F10_3_	0.953	1.788	1.223	1.208
LS20–F10	1.036	1.803	0.988	1.086
F30 ^1^	0.851	1.877	0.998	0.883

^1^ Reference formulation.

**Table 4 materials-13-02543-t004:** Healing products phases identified by XRD method for the various formulations and healing durations. Mc refers to monocarboaluminate, Hc refers to hemicarboaluminate, and Ht refers to hydrotalcite (‘x’ denotes the detection of minerals, ‘− ‘ is reported when minerals have not been detected).

Phases	MK20–BFS10			BFS20–MK10				MK20–FS10			BFS20–LS10				F30				LS20–F10		
	REF	28C-7H	28C-28H	REF	2C-28H	28C-7H	28C-28H	REF	2C-28H	28C-28H	REF	2C-28H	28C-7H	28C-28H	REF	2C-28H	28C-7H	28C-28H	REF	28C-7H	28C-28H
Ettringite	x	x	x	x	x		−	x	x	x	x	−	−	−	−	−	x	x	x	−	x
Portlandite Ca(OH)_2_	x	x	x	x	x		x	x	x	x	x	x	x	x	x	x	x	x	x	x	x
C-S-H	x	−	x	x	−	−	x	x	x	x	x	x	x	x	x	−	x	−	x	−	x
C-A-S-H	−	x	x	−	x	x	x	x	x	x	x	−	x	x	−	x	x	x	x	x	x
Calcite	−	x	x	−	x	x	x	−	x	x	x	x	x	x	−	x	x	x	x	x	x
Aragonite	−	x	x	x	x	x	x	x	−	−	−	x	x	x	−	−	x	−	x	−	−
Vaterite	−	x	x	−	x	−	−	−	−	−	−	−	x	−	−	−	x	−	−	−	x
Gibbsite Al(OH)_3_	−	−	x	x	x	x	x	x	x	x	x	x	x	x	x	x	x	x	x	x	x
Mc	x	x	x	x	x	x	x	−	−	x	x	x	−	x	−	−	−	−	x	x	x
Hc	−	x	x	−	x	x	−	−	−	x	x	x	−	x	−	−	−	−	−	−	−
Ht	−	x	x	−	x	x	−	−	−	x	x	x	−	x	−	−	−	−	−	−	−
C_4_AH_13_	−	x	x	−	x	x	x	−	x	x	x	x	x	x	−	−	−	−	x	−	x
Kuzelite	−	x	x	−	x	−	x	−	−	−	−	−	−	−	−	−	−	−	−	−	−
Siliceous hydrogaret	−	−	−	−	−	−	−	−	−	−	−	−	−	−	−	−	x	−	−	−	−
Belite C_2_S	x	−	−	x	−	−	−	x	−	−	x	−	−	−	x	−	−	−	−	−	−
Si_3_O_8_(OH)_2_	−	−	−	−	−	−	−	−		−	−	−	x	−	x	−	x	−	−	−	−
Brucite Mg(OH)_2_	−	−	x	−	x	−	x	−	x	−	−	−	−	−	−	−	−	−	−	−	−

**Table 5 materials-13-02543-t005:** Hydration phases identified in the healing products using TGA. The first values for each phase represent the percentage of a phase regarding the total mass. The second value corresponds to the percentage of the phase divided by the total percentages of all of the identified phases. Reference powder was characterized 48 h after casting.

Chemical Phases	BFS20–FS10					F30						MK20–BFS10			BFS20–MK10		
	REF	2C-14H	2C-28H	28C-7H	28C-28H	2C-7H	2C-14H	2C-28H	28C-7H	28C-14H	28C-28H	REF	2C-28H	28C-7H	REF	28C-7H	28C-28H
C-S-H	3.9	2.5	1.9	3.4	2.2	1.7	1.5	1.4	2.7	2.2	2.2	4.1	2.2	3.2	4.4	2.9	2.6
C-S-H proportion	24.5	13.4	8.2	19.1	9.1	10.3	8.3	7.3	17.3	10.7	11.3	22.3	8.3	17.1	25.4	16.4	11.2
Carboaluminate	2.9	1.5	0.8	0.4	1.7	1.6	1.2	1.1	1.6	1.4	1.7	2.4	1.9	2.8	1.5	2.3	2.4
Carboaluminate proportion	18.4	7.8	3.6	2.5	7.2	9.8	6.5	6.0	10.4	6.8	8.5	13.1	7.2	15.2	8.7	13.0	10.2
CH	1.1	3.5	3.4	1.1	1.5	3.4	3.9	3.7	1.6	1.7	2.7	2.1	0.9	0.6	2.7	0.7	1.1
CH proportion	6.7	18.3	14.8	6.4	6.5	20.3	21.1	19.4	10.6	8.4	13.9	11.4	3.5	3.3	15.7	3.7	4.8
Calcite	8.0	11.4	16.7	12.8	18.2	9.9	11.9	12.8	9.5	15.1	12.8	9.7	21.4	12.1	8.7	12.0	17.2
Calcite proportion	50.4	60.5	73.4	72.0	77.2	59.6	64.2	67.4	61.8	74.2	66.2	53.2	81.0	64.5	50.2	67.0	73.8
Total mass (mg)	91.1	58.9	64.9	51.1	64.2	58.8	58.6	61.0	74.4	55.7	53.5	70.6	63.7	55.4	80.6	54.1	56.1

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
