# Peer review of "Self-Healing Potential and Phase Evolution Characterization of Ternary Cement Blends"

_materials, 2020, doi:10.3390/ma13112543_

Round 1

Reviewer 1 Report

Manuscript Number: Materials-813055

Title: Self-healing potential and phase evolution characterization of ternary cement blends

Comments to the Author

This paper focuses on the self-healing capacity of ternary cement blends including metakaolin (MK), ground granulated blast-furnace slag (BFS), limestone (LS) and siliceous filler (F). Morphology and healing precipitation patterns were studied through optical microscopy of artificial micro-cracks and global healing product mass monitoring, while X-ray diffraction (XRD) and thermogravimetry analysis (TGA) were used to identify and quantify the produced minerals.

The authors show that the formulation containing 10% 17 MK presents the highest healing potential at an early age while the formulations containing 20% BFS with 10% LS/F displays a higher healing potential at an older age (after 28 days). Calcite, C-S-H and portlandite are the main healing products and the calcite quantity generally increases with time.

As a whole, the manuscript is quite clear and concise. The Introduction is a little bit long (82 lines of text), the analytical procedures draw on established methodologies and are properly described, and the results are interesting. However, the Discussion section appears at times repetitive and not very effective, and should be better re-organized and partly rewritten. The conclusions are supported by the analysis of the data presented and the manuscript's subject is attractive and suitable for the high-quality Materials Journal.

In my opinion, this manuscript has the potential to be published. However, I have found three major points that should be properly addressed prior to publication.

1 - The bad quality of the English style needs a deep revision by an English colleague or a professional editor in the field. The concept of a leading sentence does not seem to be present. This makes the text difficult to understand and precludes the smoothness of reading. Most paragraphs in the paper should be re-written properly, for allowing the reader to understand their content. Again, there are several inaccuracies and errors that need to be resolved. I have annotated directly on an attached pdf file some of these errors.

2 - The discussion section appears confused and prevents the reader to fully understand the research that was carried out and the obtained results. It should be rewritten.

3 – There are some problems with layout and editorial structure (repeated figures, large gaps) that need to be resolved appropriately.

To conclude, I appreciate the reading of this manuscript, but I believe that a major revision of it is necessary before being accepted by the high-quality Journal “Materials”.

Yours, sincerely,

The reviewer

Author Response

Response to Reviewer 1 Comments

Point 1: This paper focuses on the self-healing capacity of ternary cement blends including metakaolin (MK), ground granulated blast-furnace slag (BFS), limestone (LS) and siliceous filler (F). Morphology and healing precipitation patterns were studied through optical microscopy of artificial micro-cracks and global healing product mass monitoring, while X-ray diffraction (XRD) and thermogravimetry analysis (TGA) were used to identify and quantify the produced minerals.

The authors show that the formulation containing 10% 17 MK presents the highest healing potential at an early age while the formulations containing 20% BFS with 10% LS/F displays a higher healing potential at an older age (after 28 days). Calcite, C-S-H and portlandite are the main healing products and the calcite quantity generally increases with time.

Response 1: The authors would like to sincerely thank the editor in chief and the associated editor for the time spent on our manuscript and the possibility to reconsider our revised work for publishing. The authors value the comprehensive comments needed to perform this revision and the valuable suggestions made by the reviewers. Please find below our reply to the points raised and the corresponding modifications we have made in the revised manuscript to accommodate all the comments provided by the reviewers. All changes made to accommodate the referee comments are red-colored both in this document and in the manuscript (using the ‘Track changes’ mode as suggested by the journal).

Point 2: As a whole, the manuscript is quite clear and concise. The Introduction is a little bit long (82 lines of text), the analytical procedures draw on established methodologies and are properly described, and the results are interesting. However, the Discussion section appears at times repetitive and not very effective, and should be better re-organized and partly rewritten. The conclusions are supported by the analysis of the data presented and the manuscript's subject is attractive and suitable for the high-quality Materials Journal.

In my opinion, this manuscript has the potential to be published. However, I have found three major points that should be properly addressed prior to publication.

Response 2: The length of the introduction section has been reduced by around 10 lines as some minor information has been removed. The remark about the Discussion section will be discussed in Point 4.

Point 3: The bad quality of the English style needs a deep revision by an English colleague or a professional editor in the field. The concept of a leading sentence does not seem to be present. This makes the text difficult to understand and precludes the smoothness of reading. Most paragraphs in the paper should be re-written properly, for allowing the reader to understand their content. Again, there are several inaccuracies and errors that need to be resolved. I have annotated directly on an attached pdf file some of these errors.

Response 3: The manuscript has been revised by a native English speaker and some paragraphs have been revised to increase their readability, especially in the ‘Results and discussion’ section. Leading sentences have been added all the paragraphs. The inaccuracies and errors indicated in the original manuscript have been resolved accordingly.

Point 4: The discussion section appears confused and prevents the reader to fully understand the research that was carried out and the obtained results. It should be rewritten.

Response 4: Thank you for your valuable comment. The ‘Results and discussion’ part has been partly rewritten in order to be more concise and understandable. The following modifications have been done:

  • Section 3.1’ Morphology of healing products on the crack surfaces’: The different types of morphologies are addressed to the corresponded pictures and the section is extensively modified to be clearer (confusing gaps have been removed). Crystal formation inside of the cracks and crack filling is defined another way. The position of precipitation on the interior surface of the crack is precised
  • Section 3.2 ‘Self-healing mass production’: This section has been condensed and a clearer sentence to describe the effect of self-healing duration on healing potential has been added. Error bars are now included in Figure 5. The unclear sentence in section 3.2.2 has been written again.
  • Section 3.3.1 about XRD results has been extensively revised to remove repetitive and confusing sentences. Shorter sentences are proposed. The total length of this section has been reduced from 130 lines to 110 lines
  • Section 3.3.2 about TGA analysis. Subsection 3.3.2.2 about the quantification of the healing products has been partly rewritten to be clearer. More information about the establishment of table 5 is brought.

Point 5: There are some problems with layout and editorial structure (repeated figures, large gaps) that need to be resolved appropriately.

Response 5: Thank you again for this kind comment. We are deeply sorry for that. A careful revision of the format of the manuscript has been done, the repeated figures deleted and the page break causing the large gap removed.

Point 6: To conclude, I appreciate the reading of this manuscript, but I believe that a major revision of it is necessary before being accepted by the high-quality Journal “Materials”.

Response 6: Thank you for your positive reading and your suggestions. We hope that the paper has elevated to a greater level of clarity and professional contribution to the scientific outcomes.

Reviewer 2 Report

  1. Please use the paragraph format of the article to rewrite the content from line 80 to line 103 and from line 178 to line 186, etc.
  2. The content of the mortar mixture design in Table 2 is incomplete, please modify it appropriately. In addition, the mixture proportions must be clearly stated in this article in Section 2.1 and be consistent with Table 2.
  3. The experimental data are not presented with error-bars, they help in understanding the scatter in data and better analyzing the results. How many specimens were considered for each experimental data.
  4. It is confused me to calculate the porosity of the specimens by using Equation (1). Also, probably there are some typo errors in Equation 1?  What is the meaning of “The mass variation is no longer greater than 0,09g”? What is the original mass (weight) of specimens?
  1. Equation (2) needs more explanation. Why it can be as “healing potential”?
  2. The description of the research method and content is not clear, so it is difficult to determine whether the test results are reliable or repeatable, please revise it.
  3. There are several typos and grammar error in the current version. The proof-reading should be rigorously conducted before the re-submission. 

Author Response

Response to Reviewer 2 Comments

The authors would like to sincerely thank the editor in chief and the associated editor for the time spent on our manuscript and the possibility to reconsider our revised work for publishing.

The authors value the comprehensive comments needed to perform this revision and the valuable suggestions made by the reviewers.

Please find below our reply to the points raised and the corresponding modifications we have made in the revised manuscript to accommodate all the comments provided by the reviewers.

All changes made to accommodate the referee comments are red-colored both in this document and in the manuscript (using the ‘Track changes’ mode as suggested by the journal).

Point 1: Please use the paragraph format of the article to rewrite the content from line 80 to line 103 and from line 178 to line 186, etc.

Response 1: The content of the aforementioned lines has been changed according to the journal template as well as the figures and equations layout.

Point 2: The content of the mortar mixture design in Table 2 is incomplete, please modify it appropriately. In addition, the mixture proportions must be clearly stated in this article in Section 2.1 and be consistent with Table 2.

Response 2: The content of the mortar mixture in Table 2 is completed (all the masses were not initially reported in all the cells to avoid repetition). The mixture proportions and corresponded amounts are stated in Section 2.1.

Point 3: The experimental data are not presented with error-bars, they help in understanding the scatter in data and better analyzing the results. How many specimens were considered for each experimental data?

Response 3: Considering some surfaces partially nicked, corresponded errors are added to the mass measurements. This variation is explained in section 2.4 ‘Healing product conditioning and analysis’ and figure 5 is modified accordingly. As it is mentioned in the Section 2.3 ‘Artificial cracks preparation and healing conditions, for every experimental data 2 samples with dimensions of 4x4x16 cm are prepared with 30 cracks and 60 crack surfaces.

Point 4: It is confused me to calculate the porosity of the specimens by using Equation (1). Also, probably there are some typo errors in Equation 1?  What is the meaning of “The mass variation is no longer greater than 0,09g”? What is the original mass (weight) of specimens?

Response 4: Equation 1 for calculating the open porosity is given in French standard NF P18-459 supplementing the European standards. But since this value was not used in the discussion part this paragraph is removed from Section 2.4.

Point 5: Equation (2) needs more explanation. Why it can be as “healing potential”?

Response 5: More details about equation (2) are brought. The notion of ‘Healing Potential’ has been defined in equation 1 as the mass of self-healing products divided by the surface over which it precipitates. Therefore, this denomination highlights the intrinsic capacity of a material to heal.

Point 6: The description of the research method and content is not clear, so it is difficult to determine whether the test results are reliable or repeatable, please revise it.

Response 6: The description of the research method has been revised in paragraph 2.4. The new paragraph mentions explicitly the number of scratched surfaces (64) for each healing duration which guarantee reliability in our opinion. The carefully controlled artificial crack creation technique and the healing conditions described in paragraph 2.3 ensure repeatability.

Point 7: There are several typos and grammar error in the current version. The proof-reading should be rigorously conducted before the re-submission. 

Response 7: The manuscript has been revised by a native English speaker and some paragraphs have been revised to increase their readability

Round 2

Reviewer 1 Report

Dear Editor,

many thanks for reconsidering me as one of the reviewers of this study.

I'm pleased to note that the text, figures and English style have been significantly improved following the suggestions. The authors clearly have put some effort to improve the paper and the revised manuscript appears now much better than its early version. 

I think that the study can now be published in the high-quality journal Materials.

My best regards

Reviewer 2 Report

The authors have proporly interpreted the reviewer's concerns, and also revised the manuscript.